

# Short-termed changes in quantitative ultrasound estimated bone density among young men in an 18-weeks follow-up during their basic training for the Swiss Armed Forces

Michael Strässle[1,2,3], Jonas Grossmann[4,5], Patrick Eppenberger[2], Alexander Faas[6], Ivanka Jerkovic[6], Joël Floris[2], Lena Öhrström[2], Gülfirde Akgül[2], Lafi Aldakak[2], Frank Rühli[2,7], Nicole Bender[2] and Kaspar Staub[2,7]

[1] Kantonsspital St. Gallen, St. Gallen, Switzerland
[2] Institute of Evolutionary Medicine, University of Zurich, Zurich, Switzerland
[3] Medical Faculty, University of Zürich, Zurich, Switzerland
[4] Functional Genomics Center Zurich, Eidgenössische Technische Hochschule Zürich, Zürich, Switzerland
[5] SIB Swiss Institute of Bioinformatics, Lausanne, Switzerland
[6] Swiss Armed Forces, Bern, Switzerland
[7] Center for Integrative Human Physiology (ZIHP), University of Zurich, Zurich, Switzerland

Corresponding author
Kaspar Staub,
kaspar.staub@iem.uzh.ch

## ABSTRACT

**Background:** Quantitative Ultrasound (QUS) methods have been widely used to assess estimated bone density. This study aimed to assess changes in estimated bone density in association with changes in body composition, physical activity, and anthropometry.

**Methods:** We examined changes in anthropometry, body composition, and physical activity associated with changes in estimated bone mineral density (measured using quantitative ultrasound with a heel ultrasound device indicating broadband ultrasound attenuation BUA and speed of sound SOS) in a follow-up sample of $n = 73$ young men at the beginning and again 18 weeks later at the end of basic military training.

**Results:** At the end of the basic training, the subjects were on average significantly heavier (+1.0%), slightly taller (+0.5%) and had a higher fat mass (+6.6%) and grip strength (+8.6%). A significant decrease in mean physical activity (−49.5%) and mean estimated bone density calculated with BUA (−7.5%) was observed in the paired t-test. The results of the multivariable linear regressions (backward selection) show that changes in skeletal muscle mass (delta = 2nd measurement minus 1st measurement) have negative and body weight (delta) have positive association with the speed of sound SOS (delta), while fat mass (delta) and physical activity (delta) had the strongest negative associations with estimated bone mineral density (delta). In particular, we found a negative association between fat mass (delta) and estimated bone mineral density (delta, estimated with BUA).

**Conclusion:** Our study suggests that estimated bone density from the calcaneus can change within a few months even in young and mostly healthy individuals, depending upon physical activity levels and other co-factors. Further studies

including other troop types as control groups as well as on women should follow in order to investigate this public health relevant topic in more depth. To what extent the estimated bone density measurement with quantitative ultrasound is clinically relevant needs to be investigated in further studies.

# INTRODUCTION

Adverse bone health and especially osteoporosis are major public health problems, with an osteoporosis prevalence of 16% in men and 30% in women in the USA in 2017 (*Srichan et al., 2016*; *Wright et al., 2017*). In the European Union, 3.5 million cases of fragility fractures (defined as a fall from standing height or less, resulting in a fracture) were caused by osteoporosis and adverse bone health, costing an estimated 2,050 million Swiss Francs per year in Switzerland in 2010 (*Hernlund et al., 2013*). Regarding bone health, most research is conducted on older people. Less information is available on young and healthy people, although it has been proven that if prevention is necessary, it best starts at a young age. Bone density determination plays the biggest role in the study of bone health. Besides the clinical gold standard dual energy X-ray absorptiometry (DXA), quantitative ultrasound (QUS) is an established alternative to estimate bone density and predict osteoporotic fractures (*Bauer et al., 1997*; *Khaw et al., 2004*; *Chan et al., 2013*; *Hollaender et al., 2009*; *Olszynski et al., 2013*; *Esmaeilzadeh et al., 2016*) in specific study settings (*Krieg et al., 2003*; *Gluer, 2008*). Systematic reviews and meta-analyses as well as the International Society for Clinical Densitometry (ISCD) validated QUS as being a good predictor for hip and other non-vertebral fractures and as reliable as DXA (*Marín et al., 2006*; *Moayyeri et al., 2012*; *Krieg et al., 2008*). QUS is an independent predictor of fracture in men and women, even after adjusting for DXA (*McCloskey et al., 2015*), and showed the same area under the curve as DXA in calculating the risk for fracture (*Esmaeilzadeh et al., 2016*; *Cesme, Esmaeilzadeh & Oral, 2016*). Direct correlations between QUS and DXA showed a sensitivity of 70–85% and a specificity of 44–70% for detecting osteoporosis, and is therefore used as an estimated bone mineral density measurement (accuracy is not reported in these studies) (*Liu et al., 2019*; *Lee, Roh & Yoon, 2003*). Also, the US Preventive Services Task Force states that QUS at the calcaneus predicts fractures of the femoral neck, hip, and spine as effectively as DXA (*U.S. Preventive Services Task Force, 2011*).

Current clinical calcaneal bone QUS devices measure two parameters after passing through the bone: broadband ultrasound attenuation (BUA) and speed of sound (SOS). BUA and SOS provide supplementary information on the mechanical and structural properties of bone, which are distinct from bone mineral density (BMD) (*Thomsen et al., 2015*; *Hans et al., 1995*). SOS is additionally influenced by mechanical factors like elasticity and compressive strength along with structural factors like the density and architecture of trabeculae (*Guglielmi, Adams & Link, 2009*). BUA results from a combination of absorption and scattering, and reflects particularly structural properties such as bone size,

bone volume, and orientation of the trabecular network (*Hutmacher et al., 2007*; *Nicholson et al., 2000*).

Most studies investigating bone density are performed in the elderly, particularly postmenopausal women, concerning osteoporosis, while studies in young men are few. Specifically in young adults, physical activity influences bone density: high-impact sports (*e.g.*, rugby and powerlifting) lead to higher BMD (measured with DXA) than low-impact or non-weight bearing sports (such as rowing, cycling, and swimming) (*van Santen et al., 2019*). An 18-month follow-up study in gymnasts compared to controls (with 3-monthly QUS measurements) showed a continuous increase in estimated bone density (calculated with BUA), but with no change in SOS (*Daly et al., 1999*). Another study with 3-month circuit training showed an increase in SOS and BUA in young female students (*Takahata, 2018*). Similar associations have also been observed in military settings, where life circumstances (exercise, diet, *etc.*) are equal for most participants. A handful of studies in various military follow-up settings over a few months have already documented changes in bone density. For example, in young healthy recruits, a high response of bone density and remodelling microarchitecture (measured with DXA and CT scan) was observed after 8–10-weeks of physical training (*Takahata, 2018*). The same observation of increasing estimated bone density (calculated with BUA) was made after 6 months of military service in Finland, while physical training had the largest effect (*Välimäki, Löyttyniemi & Välimäki, 2006*). An investigation of a 12-week program of physical military training in the UK showed an increase in BMD (DXA), estimated bone density (BUA), and bone volume (particularly in cortical and periosteal volume), but no change in SOS (*Eleftheriou et al., 2012*). For the Swiss Armed Forces context, there is no information yet on changes in bone status during military service. In general, little is known about short-term changes in bone status of young men in Switzerland, especially about associations with changes in body composition, anthropometry, and physical activity.

This study aimed to investigate the longitudinal use of QUS in a 4 months' follow-up of a sample of basic military training recruits to assess estimated bone density in association with co-factors such as body composition, physical activity, and anthropometry.

## METHODS

The first measurement took place in March 2017 in Kloten (Canton Zurich, Switzerland); the follow-up measurements were taken 18 weeks later in July 2017 in Neunkirch (Canton Schaffhausen, Switzerland). The precise study protocol has been described elsewhere (*Sager et al., 2020*; *Beckmann et al., 2019*). The voluntary participants were Swiss male air defence recruits of the Swiss Armed Forces aged 19–23 years at the beginning and at the end of their basic military training. Typically, during the first weeks of the basic training, recruits are taught the basic knowledge for soldiers. Parallel to this, the function-related basic training begins. In the third part of the basic training, the focus is on unit training. During the training weeks, the recruits undergo a standardized program, which involves similar nutrition and physical activity levels for most recruits of a specific troop type. The air defence has in comparison to other types of troops (*e.g.*, grenadiers, infantry) a lower physical requirement.

For our study, no selection was made for socioeconomic background, regional origin, or demographic factors. Because this study was the first with follow-up in the basic training setting of the Swiss Armed Forces, and measurement times and availability of participants also depended on troop organizational factors, the sample size was not calculated. However, the power of our models was calculated *post-hoc* (see below). A total of 104 young men participated at the baseline measurements at the beginning of the basic training; 73 (70.2%) could be reassessed 4 months later. Due to splitting and relocation of subjects from the initial troop or quitting the service, we were not able to re-assess 31 subjects; thus, they were excluded from the study. The same measurement protocol and devices were used during both examinations. Participation was voluntary. Written and oral briefings were provided at the start of the study and shortly before the examination, respectively. The participants signed a detailed informed consent form. This study was approved by the Ethics Committee of the Canton of Zurich (No. 2016-01625).

## Outcome variables: estimated bone density *via* QUS

We used a calcaneal site QUS device (Sonost-3000; medical ECONET, Oberhausen, Germany) which measured the velocity of sound waves as the speed of sound (SOS in m/s) and the attenuation after passing the bone as bone ultrasound attenuation (BUA in dB/MHz). The ultrasound device was calibrated with a phantom before each use. In the range of the ultrasound measurements (between 0.3 and 0.65 MHz), theoretical calculations illustrated a linear function of attenuation dependent on frequency, and a linear positive correlation between estimated bone density (calculated with BUA) and BMD (measured with DXA) (*Liu et al., 2019*). This correlation was also observed in experimental studies (*Lee, Roh & Yoon, 2003*; *Wear et al., 2017*; *Hodgskinson et al., 1996*). A direct correlation of calcaneal quantitative ultrasound measurement (using a Sonost device) and DEXA measurements in patients with fragility fractures showed very high sensitivity and specificity. BUA showed a sensitivity of 92% and a specificity of 95% regarding the detection of osteoporosis. However, SOS showed 89% sensitivity and 85% specificity (*Moraes et al., 2011*). The manufacturer claims that the Coefficient of Variation (CV%) for SOS is 0.2 and for BUA 1.5 (the establishment of within-study CV% for this device seems to be a desideratum and must be left for follow-up studies, and other studies with the same device also refer to the manufacturer's values) (*Krieg et al., 2008*; *Jafri et al., 2020*; *Zhao et al., 2007*). In addition, QUS devices calculate the bone quality index (BQI) from SOS and BUA using the manufacturer's equation. Because of the manufacturer's custom equation which is based on SOS and BUA, there is no comparability between different devices. Therefore, we excluded the BQI from further investigations. The device was calibrated according to protocol daily before the start of measurement.

## Co-factors: body composition, anthropometric measurements, and physical activity

For the bioelectrical impedance analysis (BIA), we used a medical 8-point body composition analyser (Seca mBCA 515; Seca, Reinach, Switzerland). We measured body fat mass (%), visceral fat mass (l), skeletal muscle mass (kg), body weight (kg), and total

energy expenditure (kcal/day). Participants stood barefoot on footpads (each side with two electrodes) and held their hands on handpads (each side with two electrodes). Compared to the four-compartment measurements for determining body composition, the Seca body composition analyser correlates with 98%, which suggests an equivalent quality (*Bosy-Westphal et al., 2013*).

After the BIA measurements, we performed manual measurements of the waist circumference according to the WHO protocol; we used a handheld tape (Seca 201; Seca, Reinach, Switzerland) with stretch resistant quality and automatic retraction, at the midpoint between the lowest point of the ribcage and the highest point of the pelvis bone, while the participant stood in a relaxed upright position breathing normally (*WHO, 2000*). A standard stadiometer (Seca 274; Seca, Reinach, Switzerland) was used to determine body height, with the participants standing barefoot in a straight-up position, feet together. Body mass index (BMI, kg/m$^2$) was calculated from height and weight. BMI values were classified as underweight (<18.5 kg/m$^2$), normal weight (18.5–24.9 kg/m$^2$) and overweight (>25.0 kg/m$^2$), according to WHO guidelines. Grip strength was measured using a hand dynamometer (SH5001; Saehan, Changwon-si, Korea).

The global physical activity questionnaire (GPAQ) was used to evaluate physical activity at the beginning of the study (to assess physical activity before starting the basic training) and at follow-up (to assess physical activity during the basic training) (*Armstrong & Bull, 2006*). The questionnaire contained items about low-, mid-, and high-intensity activities during work (*e.g.*, for *low-intensity*: businessman, merchandiser; for *mid-intensity*: housekeeper, gardener, farmer; for *high-intensity*: lumberjack, construction worker, bricklayer, roofer, fitness instructor) and leisure (*e.g.*, for *low-intensity*: sedentary activities, fishing; for *mid-intensity*: casual cycling/swimming, dancing, riding, yoga, strength training, climbing; for *high-intensity*: soccer, football, athletics, aerobic, ballet, jogging, boxing, intense cycling/swimming) (*Armstrong & Bull, 2006*). The GPAQ allows to calculate the metabolic equivalents (MET) per week, which are commonly used to express the intensity of physical activities. MET is the resting metabolic rate and defined as the energy consumption of sitting quietly. One MET is equivalent to a caloric consumption of 1 kcal/kg/h. During moderate activity, the caloric consumption is four times as high as the resting metabolic rate and during vigorous activity eight times as high. To calculate the physical activity as overall energy expenditure, the sum of MET of moderate activities (MET multiplied by 4), and of MET of vigorous activities (MET multiplied by 8), was used (*Armstrong & Bull, 2006*).

The participants were also asked about their smoking and sports habits. Accordingly, answers were categorized into groups, athletes *vs.* non-athletes (regular sportive activities in leisure time *vs.* no sportive activities in leisure time), and smokers *vs.* non-smokers (number of cigarettes per day > 0 *vs.* number of cigarettes per day = 0).

## Statistics

Outcome variables (BUA, SOS) and co-factors (body composition measures, anthropometrics, and physical activity levels) at baseline and follow-up were reported as mean values ± standard deviation (SD) for the full sample as well as for athletes *vs.*

**Table 1 Descriptive characteristics of subjects at beginning and follow-up with *p*-values of changes.**

| Characteristics | Start ± SD | Follow-up ± SD | Delta | *p*-value |
|---|---|---|---|---|
| Age (years) | 20.5 ± 1.0 | | | |
| Bone measurements | | | | |
| SOS (m/s) | 1,545.3 ± 15.5 | 1,543.5 ± 14.6 | −1.9 | 0.145 |
| BUA (dB/MHz) | 87.8 ± 16.8 | 81.2 ± 16.1 | −6.6 | <0.001 |
| Anthropometrics | | | | |
| Gripstrength (kg) | 47.8 ± 9.5 | 51.9 ± 9.2 | +4.1 | <0.001 |
| BMI (kg/m²) | 23.0 ± 3.3 | 23.2 ± 3.1 | +0.2 | 0.122 |
| Fat mass (%) | 13.7 ± 7.1 | 14.6 ± 6.9 | +0.9 | 0.003 |
| Fat free mass (kg) | 62.46 ± 6.7 | 62.47 ± 6.2 | +0.01 | 0.964 |
| Skeletal muscle mass (kg) | 30.67 ± 3.7 | 30.71 ± 3.3 | +0.04 | 0.740 |
| Waist circumference (cm) | 0.7999 ± 0.1 | 0.8004 ± 0.1 | +0.0005 | 0.890 |
| Weight (kg) | 73.2 ± 12.4 | 73.9 ± 11.6 | +0.7 | 0.041 |
| Height (m) | 1.78 ± 0.1 | 1.79 ± 0.1 | +0.01 | <0.001 |
| Visceral adipose tissue (l) | 0.73 ± 0.8 | 0.72 ± 0.6 | −0.01 | 0.771 |
| Physical activity (MET) | 7,547.9 ± 6,570.6 | 3,815.3 ± 3,596.6 | −3,732.6 | <0.001 |

Note:
Descriptive characteristics of the participants at baseline and 18 weeks later at follow-up (*n* = 73). Subjects who could not be reassessed in follow-up were excluded from the study (*n* = 31) and are not shown in this table. SD, standard deviation. *p*-value from the paired t-test with level of significance <5%. SOS, speed of sound; BUA, bone ultrasound attenuation; BMI, body mass index; MET, metabolic equivalents per week.

non-athlete and smokers *vs*. non-smokers. Differences between means were tested using paired t-tests. The symmetry of the main variable distributions of each parameter was visually assessed using histograms.

The individual longitudinal changes in the outcome variables and co-factors were calculated as deltas (delta = 2nd measurement minus 1st measurement). First, Pearson correlation coefficients were calculated between the individual deltas for the outcome variables BUA and SOS on the one hand and all cofactors on the other. Second, to see which co-factors delta were most associated with delta in the outcome variables (BUA and SOS), we used stepwise backward selection and calculated multi-variable linear regression models which included anthropometrics, body composition, physical activity, and smoking status as independent variables. We also calculated *post-hoc* power of our models using the *pwr* package.

All statistical analyses were performed using R version 4.1.2 (*R Core Team, 2021*).

## RESULTS

The descriptive statistics of the outcome variables and co-factors are given in Table 1. At baseline, the age of the participants ranged from 18.8 to 23.9 years (mean 20.5, SD 1.0), body height ranged from 1.66 to 1.94 m (mean 1.78, SD 0.07), weight ranged from 50.6 to 104.7 kg (mean 73.2, SD 12.4), and BMI ranged from 16.4 to 30.1 kg/m² (mean 23.0, SD 3.3). Eighteen subjects (24.7%) were overweight (BMI >= 25.0 kg/m²), and three subjects (4.1%) were underweight (BMI < 18.5 kg/m²). At follow-up 18 weeks later, 20 of 73 subjects (27.4%) were overweight, while the prevalence of underweight was unchanged.

**Table 2 Correlation values between delta of SOS and BUA and delta of variables.**

| Variable (delta) | | SOS | BUA |
|---|---|---|---|
| Gripstrength (kg) | r | **0.06** | −0.03 |
| | p | **0.032** | 0.162 |
| BMI (kg/m$^2$) | r | −0.05 | −0.03 |
| | p | 0.063 | 0.113 |
| Fat mass (%) | r | 0.05 | **−0.13** |
| | p | 0.070 | **0.002** |
| Fat free mass (kg) | r | **−0.21** | 0.01 |
| | p | **<0.001** | 0.363 |
| Skeletal muscle mass (kg) | r | **−0.24** | 0.01 |
| | p | **<0.001** | 0.360 |
| Waist circumference (cm) | r | **−0.05** | −0.05 |
| | p | **0.048** | 0.067 |
| Weight (kg) | r | −0.05 | −0.03 |
| | p | 0.056 | 0.119 |
| Height (m) | r | 0.00 | −0.01 |
| | p | 0.818 | 0.413 |
| Visceral adipose tissue (l) | r | −0.03 | −0.03 |
| | p | 0.144 | 0.143 |
| Physical activity (MET) | r | 0.00 | **−0.84** |
| | p | 0.685 | **0.013** |

**Note:**

Pearson correlation coefficients between deltas of SOS and BUA and deltas of co-factors. r, Pearson correlation coefficient (statistically significant results are highlighted in bold). $p$-value with level of significance <5%. SOS, speed of sound; BUA, bone ultrasound attenuation; BMI, body mass index; MET, metabolic equivalents per week.

All variables appeared to be symmetrically distributed. There were 43 subjects in the athletes group (58.9%) and 30 subjects in the non-athletes group (41.1%). In the smoker group ($n$ = 34, 46.6%), two stopped smoking during the study period. In the non-smoker group, ($n$ = 39, 53.4%), two started smoking.

The comparison of mean values between baseline and follow-up is shown in Table 1. The means for weight, height, fat mass, and grip strength significantly increased during basic training, while mean BUA and physical activity significantly decreased during the same period. No significant changes were observed in mean BMI, fat-free mass, skeletal muscle mass, waist circumference, visceral adipose tissue, and SOS. When comparing the athletes *vs*. non-athletes groups as well as the smoker *vs*. non-smoker groups, the same patterns were observed (Tables S1 and S2).

When the individual changes (delta) in selected outcome variables and co-factors are plotted against their baseline value at baseline (Fig. S1), it becomes apparent that most variables (except height) showed regression to the mean-like behaviour with an approximation to the mean: High values at the beginning dropped and low values at baseline increased during basic training. Pearson correlation coefficients between deltas in the outcome variables (SOS and BUA) on the one hand and deltas in the co-factors on the other hand are reported in Table 2. For individual changes in SOS, a weak positive

**Table 3 Multiple regression results according to bone parameters SOS and BUA.**

| Parameter | Independent variables | β coefficient | p value (variable) | $R^2$ (model) | p value (model) |
|---|---|---|---|---|---|
| SOS | Skeletal muscle mass (kg) | −6.01 | <0.001 | 0.32 | <0.001 |
| | Weight (kg) | 0.93 | 0.086 | | |
| BUA | Fat mass (%) | −1.74 | <0.001 | 0.22 | <0.001 |
| | Physical activity (MET) | −0.0005 | 0.006 | | |

Note:
p-value with level of significance <5%. $R^2$, goodness of fit; SOS, speed of sound; BUA, bone ultrasound attenuation; MET, metabolic equivalents per week.

correlation was observed with grip strength ($R^2 = 0.06$, $p = 0.03$), whereas weak negative correlations were found with skeletal muscle mass ($R^2 = 0.25$, $p = <0.001$) and waist circumference ($R^2 = 0.05$, $p = 0.048$). For individual changes in BUA, there were weak negative correlations with fat mass ($R^2 = 0.13$, $p = 0.002$) and physical activity ($R^2 = 0.08$, $p = 0.013$).

Results from the multivariable linear regressions (backwards selection) are reported in Table 3. For the model with delta of SOS as dependent variable, delta in skeletal muscle mass ($\beta$-coefficient −6.01, $p < 0.001$) and delta in body weight ($\beta$-coefficient +0.93, $p = 0.086$) were the remaining co-factors, the model explained nearly one-third of the variation in delta of SOS ($R^2 = 0.32$). For the model with delta in BUA as dependent variable, delta in fat mass ($\beta$-coefficient −1.74, $p < 0.001$) and delta in physical activity ($\beta$-coefficient −0.0005, $p = 0.006$) were the remaining co-factors, the model explained nearly a quarter of the variation in delta of BUA ($R^2 = 0.22$). The very low $\beta$-coefficient of physical activity was due to higher absolute values (mean at start = 7,547.9) than fat mass (mean at start = 13.7). The *post-hoc* calculated power of the two models was 0.998.

## DISCUSSION

In this study, we analysed changes in the estimated bone density at the calcaneus (calculated with BUA) and their associations with the co-factors like anthropometry, body composition, and physical activity in a short-term follow-up setting (18 weeks) during basic military training in a sample of 73 young Swiss men. At the end of their basic training, the participants were by average heavier, slightly taller, and had higher fat mass and grip strength. A significant decrease in mean physical activity and estimated bone density (calculated with BUA) was observed. Generally, a regression to the mean-like change of individual differences was visible, except for height. Weakly negative associations were found between deltas (delta = 2nd measurement minus 1st measurement) in estimated bone density and deltas in fat mass as well as physical activity. As has been proven in various studies (see introduction), the QUS measurement can also show a change in the estimated bone density over a short period of 3 months (*Daly et al., 1999*; *Takahata, 2018*).

Several studies have investigated the associations of anthropometry and QUS in children, young and elderly adults, particularly women (*Lavado-Garcia et al., 2012*; *Szmodis et al., 2019*; *Babaroutsi et al., 2005*). In children and men, age, weight, height, BMI, and fat-free mass were positively correlated with SOS and BUA, while fat mass and waist

circumference were correlated with BUA. Vigorous physical activity correlated positively with SOS and BUA, but dietary factors showed no association (*Lavado-Garcia et al., 2012*; *Szmodis et al., 2019*; *Babaroutsi et al., 2005*). In our study, an increase in height during basic training was visible due to growth. Physical growth can be observed until the age of 24 (*Bogin, 2020*). The general increase in grip strength at the end of basic training could be explained by increased physical load caused by manual work (*e.g.*, repeated packing, carrying, or putting on and taking off the heavy backpacks).

At the end of basic training, an increase in weight and fat mass and a decrease in physical activity were observed. No changes in fat-free mass or skeletal muscle mass were observed; thus, it can be assumed that the increase in weight is due to increased fat mass. Maybe, recruits had less physical activity during the basic training than before, which can explain the gain in weight and fat mass. Furthermore, changes in eating habits (*e.g.*, regular breaks for subsistence) and/or a higher physical/mental stress level could have negative effects on weight and fat mass (*Stefanaki et al., 2018*).

After the 18-week follow-up, a change in the estimated bone density is visible with a significant decrease in the attenuation of the ultrasound (BUA). Interestingly, the regressions revealed a weak association between changes in BUA and changes in fat mass, which could indicate that increase in fat mass—alongside with decreased physical activity—during the 18 weeks of basic training already harmed the estimated bone density. The negative influence of fat mass on bone metabolism has already been documented several times (*Zhao et al., 2007*; *Kim et al., 2012*; *Riggs, Khosla & Melton, 2002*; *Lee et al., 2008*). It is known that greater mechanical stress (with weight gain) on the bone can lead to an adaptation and increase in bone mass, as it has been shown for high-impact sports but less so for walking alone (*van Santen et al., 2019*; *Hind & Burrows, 2007*; *Nikander et al., 2010*; *Heinonen et al., 2000*; *MacKelvie et al., 2002*; *Fuchs, Bauer & Snow, 2001*; *Gunter et al., 2008*). However, it is also known that visceral and subcutaneous adipose tissue negatively affects bone remodelling *via* inflammatory factors such as the upregulation of the nuclear factor κB ligand, which leads to a stimulation of the osteoclast activity and thus to bone resorption (*Riggs, Khosla & Melton, 2002*; *Lee et al., 2008*; *Campos et al., 2012*; *Gilsanz et al., 2009*; *Russell et al., 2010*). Other negative influences of adipose tissue on bone density are also known. The amount of marrow adipose tissue, insulin resistance, increased visceral adipose tissue, and increased intrahepatic lipid measurements are negatively associated with bone density in trabecular bone (*de Araújo et al., 2020*). Also, visceral adipose tissue and estradiol deficiency independently and additively affect bone mineral density, revealing an unexpectedly high prevalence of osteopenia in middle-aged men with metabolic syndrome (*Ornstrup et al., 2015*). Our results imply that the weight gain, which is mainly due to the increase in fat mass, already has a weak negative association with estimated bone density (calculated with BUA) in the short period of 18 weeks.

Overall, no statistical change in the speed of sound was found in the follow-up. Interestingly, there are also no significant changes in SOS in various other studies, as already mentioned in the introduction (*Daly et al., 1999*; *Eleftheriou et al., 2012*). The reasons for this are not yet known, but this means that BUA is more meaningful. However, there was a positive association between sound velocity and increase in grip
strength, but a negative association with skeletal muscle mass and waist circumference. As discussed above, grip strength could indicate increased physical activity during baseline training, which could result in an increase in estimated bone density. However, there is a very low correlation here ($R^2 = 0.06$), which is why this association should be treated with caution. Regarding the negative association between SOS and skeletal muscle mass, it can be seen from the appendix that subjects with initial high skeletal muscle mass lose a lot of muscle mass during basic training. The same phenomenon could be observed for physical activity. This may be a protective effect of skeletal muscle mass, which is why subjects with the highest loss of muscle mass still have higher SOS and therefore estimated bone density at 18 weeks. However, this is only a theory, as far as we know there is no literature on this. Regarding the negative association between SOS and waist circumference, it is known that waist circumference correlates with visceral fat tissue (*Berentzen et al., 2012*), which is why the negative influence of fat tissue on estimated bone density can also apply here (as already described above).

Our study had several strengths and limitations. The strengths include the homogeneous sample of young men, the well-controlled environment, and the uniform amount and type of physical activity and nutrition during basic training. The main limitation of the study is the lack of a control group. It is known that air defence recruits have less demand for activities (marches, runs, inactivity per day) than other units in the Swiss Armed Forces (*Wyss, Scheffler & Mäder, 2012*). To better understand the circumstances of these highlighted changes in estimated bone density and associations with co-factors on the group and individual levels, a larger study design with one or more control groups (*e.g.*, other troop types) would be desirable. Further limitations include the shortcoming of distinguishing between high-and low-impact activities/sports in the GPAQ. A questionnaire may be answered subjectively; therefore, it has limited reliability. The examination battery did only include grip strength as measurement of physical fitness, and a comprehensive assessment of nutrition and diet in a longer and thus more time-consuming questionnaire could not be performed in the current setting. Information about alcohol consumption should also be included in future studies, as lifestyle behaviour can change during basic training and there is a known negative association with bone density. Furthermore, BIA is not the gold standard for measuring body composition. However, due to time constraints within the army setting, we were not able to perform more time-consuming (*e.g.*, multiple quantitative ultrasound measurements of the same subject) and invasive measurements. Similarly, waist circumference could be measured only once per measurement slot because of time constraints. By appointing the same experienced researcher, we excluded inter-observer bias; however, an intra-observer bias could not be excluded. Our sample size was limited owing to the setting reasons. As the sample was homogeneous, the internal validity of the study was given, but the external validity was limited. Other quality control measures, such as determination of heel thickness and repeated measurements for subjects, could not be collected due to time constraints in our study setting. The measurement of the existing temperature during the examinations was also not recorded. To verify the results from our study, a prospective randomized study comparing low and high impact activities as well as a detailed survey of

eating and nutritional habits (particularly alcohol intake) during basic training should be carried out. More studies including more age groups and both sexes should be performed. To what extent the estimated bone density measurement with quantitative ultrasound is clinically relevant needs to be investigated in further studies, since there were no stress fractures in our cohort group.

## CONCLUSION

The loss of physical activity of the recruits during the basic training suggests a low level of physical and athletic request in this type of troop, which presumably led to weight gain. The weight gain, which is mainly due to the increase in fat mass, already shown a negative association with estimated bone density (calculated with BUA) in the short period of 18 weeks. As shown in other studies, the force exerted on the bones while walking is insufficient to strengthen the bone. For troop types with a low or medium load pattern, we recommend additional physical exercise during basic military training, such as a daily standard jump program to strengthen the bone.

## ACKNOWLEDGEMENTS

This article was part of the Dr. med. thesis project of Michael Strässle. The authors are especially thankful to Andreas Stettbacher (Chief Medical Surgeon), Franz Frey, Martino Ghilardi, and Marco Müller from the Swiss Armed Forces for their tremendous (logistic) support. We also thank the former IEM collaborators Nikola Koepke, Anne Lehner, Claudia Beckmann, and Nakita Frater for helping to collect the data.

### Funding
The work has been supported by the Mäxi Foundation Zürich (Grantee Frank Rühli). The funders had no role in study design, data collection and analysis, decision to publish, or preparation of the manuscript.

### Grant Disclosures
The following grant information was disclosed by the authors:
Mäxi Foundation Zürich (Grantee Frank Rühli).

### Competing Interests
The authors declare that they have no competing interests. Alexander Faas and Yvanka Jerkovic are employed by Swiss Armed Forces.

### Author Contributions
- Michael Strässle conceived and designed the experiments, performed the experiments, analyzed the data, prepared figures and/or tables, authored or reviewed drafts of the article, and approved the final draft.
- Jonas Grossmann performed the experiments, analyzed the data, prepared figures and/or tables, authored or reviewed drafts of the article, and approved the final draft.

- Patrick Eppenberger performed the experiments, authored or reviewed drafts of the article, and approved the final draft.
- Alexander Faas conceived and designed the experiments, authored or reviewed drafts of the article, and approved the final draft.
- Ivanka Jerkovic conceived and designed the experiments, authored or reviewed drafts of the article, and approved the final draft.
- Joël Floris performed the experiments, authored or reviewed drafts of the article, and approved the final draft.
- Lena Öhrström performed the experiments, authored or reviewed drafts of the article, and approved the final draft.
- Gülfirde Akgül performed the experiments, authored or reviewed drafts of the article, and approved the final draft.
- Lafi Aldakak performed the experiments, authored or reviewed drafts of the article, and approved the final draft.
- Frank Rühli conceived and designed the experiments, performed the experiments, authored or reviewed drafts of the article, funding, and approved the final draft.
- Nicole Bender conceived and designed the experiments, performed the experiments, analyzed the data, authored or reviewed drafts of the article, supervision, and approved the final draft.
- Kaspar Staub conceived and designed the experiments, performed the experiments, analyzed the data, authored or reviewed drafts of the article, supervision, and approved the final draft.

## Human Ethics

The following information was supplied relating to ethical approvals (*i.e.*, approving body and any reference numbers):

Participation in this study was voluntary. The participants signed a detailed informed consent form after oral and written briefings. This study was approved by the Ethics Committee of the Canton of Zurich (No. 2016-01625).

## Data Availability

The raw measurements are available in the Supplemental File.

## Supplemental Information

Supplemental information for this article can be found online at http://dx.doi.org/10.7717/peerj.15205#supplemental-information.

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
