# Peer review of "Short-termed changes in quantitative ultrasound estimated bone density among young men in an 18-weeks follow-up during their basic training for the Swiss Armed Forces"

_PeerJ, doi:10.7717/peerj.15205_

## Round 0.1 · original submission · Minor Revisions

Please address reviewers' concerns, especially reviewer 2, and revise the manuscript accordingly.

·

Basic reporting

Authors innovatively used the quantitative ultrasound (QUS) method to assess changes in bone mineral density in young adults over a short period of time. Overall, the manuscript is well written and I suggest that it could be published after minor revisions.

Experimental design

well done

Validity of the findings

well done

Additional comments

1.Authors may consider streamlining the abstract to make it more focused on the purpose of the study.
2.It would have been better if the authors had collected other baseline characteristics of the subjects. This is just my suggestion and not forced.
3.The authors' statement in the conclusion of the abstract that QUS appears to be a valuable tool for assessing BMD at follow-up is based on conclusions drawn from previous studies and not from the present study. The authors should avoid giving the reader the impression that the study is a measure of the utility of QUS, since in the present study, QUS is a means and not an outcome.
4.The authors concluded that the implementation of QUS as a screening tool in clinical practice remains limited, mainly due to the lack of standardization of QUS measurement. Could the authors then discuss how to address this issue?

·

Basic reporting

The authors investigated the changes in quantitative ultrasound indices and their predictors after 18 weeks of army training. They found that changes in SOS were negatively predicted by changes in skeletal muscle mass and positively with changes in weight. Changes in BUA were negatively predicted by changes in fat mass and physical activity. The writing is clear and easy to understand, but I suggest the reporting of association should be accompanied by the direction of the association (negative/positive). The results are clearly illustrated using tables, but I suggest the calculated delta values for each variable to be added in Table 1.

Experimental design

While the findings of longitudinal changes in QUS indices are much appreciated, I have several comments on the reporting of QUS methods:
Bone health measurement by QUS is not equivalent to bone density, so these two concepts should not bed used interchangeably in the text.
To determine the change is not due to random error, the authors should report the performance of the machine in their setting as short-term in vivo coefficient variation. Without this information, it is hard to convince the reader the change is meaningful.
Other quality control measures, such as measuring at a constant temperature, determination of heel thickness, and repeated measurements for subjects should be reported.
What is the agreement between this QUS machine with DXA?
Actually, SOS and BUA from machines of different manufacturers may not be comparable as well. So the decision not to report BQI does not make sense. Suggest adding on data of BQI.
Was alcohol intake considered a predictor in this study?

Validity of the findings

The validity of the findings depends on whether the authors can demonstrate the change in QUS indices is clinically meaningful.

Additional comments

Minor comments:
Table 1: add the word "mean" in the label of the column.
Table 3: please indicate clearly that it is "delta" for both dependent and independent variables.

Reviewer 3 ·

Basic reporting

Literature and reference citing are required in some places. Check my additional comments.

Experimental design

no comments

Validity of the findings

Improvement required. Please check my comments in the Additional comment section

Additional comments

Line 227: Regarding the significant positive correlation: here the correlation R-squared value is 0,06. This is almost No correlation. Authors need to explain convincingly why they are considering it a "significant positive correlation"

Line 228: Regarding the negative correlation between Skeletal mass and waist circumference: R-squared values are: -0,24 and -0,05.
This is far away from being negatively correlated.
Please explain, why the authors consider this a significant negative correlation.

Line 249, 250: Here I am assuming the author is talking about Figure 3. The correlation value for Fat mass is 0.129 and for physical activity is 0.084.
These values are far from being considered Negative associations.
Please explain your argument in a convincing manner

Line 220: Table 1,2 and Appendix Table 1,2 are confusion.
In the final manuscript, please put all in the main manuscript as Table 1,2,3,4

Line 57, 58: Please cite relevant literature.

Line 85 and 86: Please cite this claim using multiple citations

Line 158: Please cite WHO protocol

Line 161: “… breathing normally (2000)“ It’s not clear what this 2000 number means

Line 165: Please cite WHO guidelines

Line 166: describe the details on the Dynamometer (model name, city, country)

Line 209 and 210: m2 should be written using superscript.

Line 218: What about Visceral adipose tissue? Please add.

Line 235 and Line 237: Please put r2 as r2 in superscript.

Line 243: with the co-factors LIKE anthro...

Line 250,251: Please cite this study.

Appendix figure 1 E: Please put the R square and p-value box on the right-hand upper side.

---

## Round 0.2 · Minor Revisions

Dear Authors,

Thank you very much for the revised manuscript. After reviewing the revised version, I am glad to inform you the submission is almost ready to be accepted for publication.

Please address the minor points from Reviewer 2.

·

Basic reporting

More careful revision would be appreciated. Please refer to my specific points below.

Experimental design

The design is adequate.

Validity of the findings

Manufacturer-reported coefficient of variance tends to be lower than in-house values. I really hope that the authors take the initiative to establish the CV values for the QUS machine. These values are critical in their study.
Although it may be too demanding to request the author to establish the correlation between QUS indices of their machine with BMD of DXA, I hope the authors can quote published studies that used the same brand of the machine as a reference (Sonost 3000).

Additional comments

Line 200-202: Multiple blank brackets without values.
Discussion: The effects of fat mass on bone health are under-discussed. Apart from inflammation, the should also mention the role of adipose tissue in secreting bone-active adipokines, sequestering sex hormones and vitamin D etc.
Discussion: The relationship between SOS and lean mass and physical activity is not discussed. I understand some previous studies have been quoted in the introduction, but in the discussion, the authors should also postulate how these changes occur and why it is dissimilar from previous findings (if needed).

Reviewer 3 ·

Basic reporting

Fine

Experimental design

Fine

Validity of the findings

Fine

Additional comments

NA

---

## Round 0.3 · Minor Revisions

Dear Authors,

Please address the attached comments from the Section Editor.

---

## Round 0.4 · accepted · Accept

Thank you very much for addressing all comments! And the submission is accepted for publication.